

# TOAR-classifier v2: A data-driven classification tool for global air quality stations

Ramiyou Karim Mache[1], Sabine Schröder[1], Michael Langguth[1], Ankit Patnala[1], and Martin G. Schultz[1]

[1]Jülich Supercomputing Centre, Forschungszentrum Jülich, 52425 Jülich, Germany

**Correspondence:** Ramiyou Karim Mache (k.mache@fz-juelich.de)

**Abstract.**

Accurate characterization of station locations is crucial for reliable air quality assessments such as the Tropospheric Ozone Assessment Report (TOAR). This study introduces a machine learning approach to classify 23,974 stations in the unique global TOAR database as urban, suburban, or rural. We tested several methods: unsupervised K-means clustering with three clusters, and an ensemble of supervised learning classifiers including random forest, CatBoost, and LightGBM. We further enhanced the supervised learning performance by integrating these classifiers into a robust voting model, leveraging their collective predictive power. To address the inherent ambiguity of suburban areas, we implement an adjusted threshold probability technique. Our models, trained on the TOAR station metadata, are evaluated on 1,000 unseen data points. K-means clustering achieves 70.03% and 71.53% accuracy for urban and rural areas respectively, but only 26.36% for suburban zones. Supervised classifiers surpass this performance, reaching over 84% accuracy for urban and rural categories, and 62-65% for suburban areas. The adjusted threshold technique significantly enhances overall model accuracy, particularly for suburban classification. The good separation of our model is confirmed through evaluation with NOx and PM2.5 concentration measurements, which were not included in the training data. Furthermore, manual inspection of 25 individual sites with Google maps reveals that our method provides a better label for the station type than the labels that were reported by data providers and used in the model evaluation. The objective station classification proposed in this paper therefore provides a robust foundation for type-specific air quality assessments in TOAR and elsewhere.

## 1 Introduction

Ozone in the troposphere plays a crucial role in human and environmental health, (Post et al., 2012; Griffiths et al., 2021). As a significant atmospheric pollutant and greenhouse gas, ozone profoundly impacts air quality and contributes to the dynamics of climate change (Madronich et al., 2023; Orru et al., 2013). Accurate ozone monitoring data is essential for shaping public health policies and ecological regulations. Modern data infrastructures can be used to provide atmospheric scientists with the necessary metrics to quantify ozone's impact on climate, human health, and vegetation (Gaudel et al., 2018; Fleming et al.; Mills et al., 2018; Teakles et al., 2017; Cooper et al., 2014; Monks et al., 2015; Schultz et al., 2015). In 2021, the International Global Atmospheric Chemistry project (IGAC) launched the second phase of the Tropospheric Ozone Assessment Report (TOAR-II) to undertake a comprehensive review of the global distribution and trends of tropospheric ozone. A key



accomplishment of TOAR-II is the development of a new terabyte-scale relational database of surface ozone observations and related variables. This database includes hourly measurement data and enriched metadata from 1970 to 2023, collating information from over 20,000 measurement sites worldwide through collaboration among multiple data centers and individual researchers (Schröder et al., manuscript in preparation). The new TOAR-II database replaces and extends the first TOAR

database that has been described in Schultz et al. (2017a). Ozone levels exhibit significant regional variations and distinct patterns across different pollution environments. For example, urban environments with large ozone precursor emissions can exhibit "zero ozone" (i.e., ozone at sub-nmol fractions) situations and very large variability, while concentrations in rural areas tend to be smoother (Zhou et al., 2022; Schultz et al., 2017b). To accurately assess the ozone situation at individual locations and interpret ozone trends across the globe, it is therefore important to characterize measurement sites in a globally

consistent and objective manner. While many measurement networks provide information about the station location or "type of station" and "type of station area", this metadata information is inconsistent between regions and error-prone as it involves some subjective judgement.

In the first assessment of TOAR Schultz et al. (2017a) pioneered a new way to classify stations in a globally uniform way based on a set of Earth Observation (EO) datasets that have been processed at the station locations. This method used manually

selected threshold values of station altitude, population density, nighttime light intensity, $NO_2$ column density, and NOx emissions to characterize stations as urban or rural. While this approach provided a useful distinction between "clearly urban" and "clearly rural" sites, it fell short of classifying all sites (almost half of the stations remained unclassified) and was criticized for lack of an objective definition of the threshold values. In this study, we propose a new machine learning (ML) approach to develop a more advanced and unbiased classifier using similar objective metadata from the TOAR-II database. Our primary

objective is to create a machine learning model that classifies stations in the TOAR database as urban, suburban, or rural. We implement and compare two methodologies: unsupervised learning using K-means clustering, and supervised learning classifiers such as random forest, CatBoosting, and LightGBM. In supervised learning, a subset of station characteristics is known and used to train the classifiers which are then used to predict the class of unclassified station locations. The supervised models are evaluated individually and after applying a robust voting method. Furthermore, an adjusted threshold technique

is applied to enhance the separation of suburban stations. The next section presents the data and methodology, followed by Section 3, which focuses on the results and discussion. A general conclusion wraps up the paper.

## 2 Data and methods

This section provides an overview of our data sources and methodology. We begin by introducing the TOAR-II database and the station metadata that are used as inputs of the ML models. We then detail our data preparation process, including

preprocessing, feature engineering, and feature selection, all critical steps in preparing data for ML models. Finally, we present a concise summary of the ML models employed in this study and the evaluation metrics used to assess their performance.



## 2.1 TOAR-II database and station metadata

Developed in the context of TOAR phase II, the TOAR-II database stands as one of the world's largest collections of near-surface ozone measurements and related information. The database can be accessed through web services which provide
a comprehensive suite of ozone-related data products including standard statistics, health and vegetation impact metrics, and trend information (https://toar-data.fz-juelich.de/). The TOAR-II database includes extensive information describing the locations of air quality measurement stations based on pollution-relevant properties. These properties are extracted from EO data and stored as station metadata in the database. This metadata offers contextual information about the measurement site, enabling station location characterization. Table 1 below summarizes all metadata used in this study including references to the data origin.

**Table 1.** Station metadata in the TOAR database used in this work.

| Variable name | Description | Type |
|---|---|---|
| lon, lat | longitude, latitude represent the geographical coordinates of station. We did not use these coordinates as predictors in the machine learning model | Numeric |
| area_code | Unique code of the station in TOAR database | String |
| altitude | altitude of the station location in meter (m) | Numeric |
| mean_topography_srtm_alt_90m_year1994 mean_topography_srtm_alt_1km_year1994 | mean value within 90m and 1km of relative altitude of the year 1994. data source: NASA Shuttle Radar Topographic Mission (SRTM) (Jarvis et al., 2008) | Numeric |
| max_topography_srtm_relative_alt_5km_year1994 min_topography_srtm_relative_alt_5km_year1994 stddev_topography_srtm_relative_alt_5km_year1994 | maximum, minimum, and standard deviation of the relative altitude within a radius of 5km around the station in 1994 data source: NASA Shuttle Radar Topographic Mission (SRTM) (Jarvis et al., 2008) | Numeric |
| climatic_zone_year2016 | climate zone of the year 2016. Provides information about climatic conditions at a location including whether it tends to be hot or cold, humid or dry, or exhibits a tropical climate data source: University of East Anglia Climatic Research Unit (Harris and Jones, 2017) | Category |
| mean_stable_nightlights_1km_year2013 mean_stable_nightlights_5km_year2013 max_stable_nightlights_25km_year2013 max_stable_nightlights_25km_year1992 | average and maximum nighttime light value of the years 1992, and 2013 in 1km, 5km, and 25km around the station location. The values in this data set represent a brightness index ranging from 0 to 63. data source: NOAA National Centers for Environmental Information (NCEI) (Kroehl, 1982) | Numeric |
| mean_population_density_250m_year2015 mean_population_density_5km_year2015 max_population_density_25km_year2015 mean_population_density_250m_year1990 mean_population_density_5km_year1990 max_population_density_25km_year1990 | Average and maximum population density of the years 1990, and 2015 in 250m, 5km, and 25km radius around the station location data source: The European Commission, Joint Research Centre, (Florczyk et al., 2019) | Numeric |
| mean_nox_emissions_10km_year2015 mean_nox_emissions_10km_year2000 | Average annual NOx emission of the years 2000 and 2015 in a 10km radius around the station location data source: Copernicus Atmosphere Monitoring Service (Granier et al., 2019) | Numeric |
| type_of_area (target) | Characterization of station location (urban, suburban, rural, or unknown) reported by the data providers of the TOAR database. This variable is not used in K-means clustering, and the known part, i.e stations labelled as urban, suburban, rural, are employed to train supervised classifiers. | Category |




Some metadata, such as station coordinates, are provided by many air quality agencies and scientific institutions that contribute data to TOAR. The other metadata elements listed in Table 1 stem from the following EO datasets, which were downloaded from the respective provider sites. A special web service called Geospatial point extraction and aggregation service (GeoPEAS) has been developed to compute the aggregate information from the original gridded products. Information

about the EO datasets used in GeoPEAS can be found on https://toar-data.fz-juelich.de/api/v2/#stationmeta. After the metadata extraction with GeoPEAS, all metadata are available as lists of key-value pairs with the keys corresponding to the variable names in Table 1. For further processing, this data was collected into one table with the keys as data columns and the individual stations as rows.

## 2.2 Data pre-processing and feature selection

The first step in our data prepossessing pipeline consists of cleaning the dataset. This involved removing all duplicate data points, replacing values of -999.0 with NaN to denote missing values, and eliminating rows where all metadata information is missing. To ensure consistency, we filtered out rows with inconsistent values, such as negative population density or negative maximum stable lights. In total, this eliminates 1,596 stations out of 23,974 stations. For handling missing altitude data, we fill these with mean_topography_srtm_alt_90m_year1994. Other missing values are estimated using the regression

iterative imputer, Rubinsteyn and Feldman (2016). Categorical variables are encoded using OneHotEncoder from scikit-learn, Pedregosa et al. (2011b). Additionally, we applied a Box-Cox transformation to the NOx emissions data to normalize its distribution. We then applied a robust scaler, Pedregosa et al. (2011b) to the entire dataset, which proved more effective for this task compared to alternative scaling methods such as standard or min-max scaling. This scaling approach helps mitigate the impact of outliers and ensures consistent feature ranges. In the feature selection process, we prioritized variables containing the

most recent available information, ensuring our model utilizes the most up-to-date data for classification. Notably, geographical coordinates (longitude-latitude) and station codes are excluded from the machine learning models to prevent overfitting to geolocation.

After preprocessing, the dataset consists of 22,378 rows. For the K-means clustering analysis, we allocated 22,378 samples for model training and reserved 1,000 samples for testing purposes. The training of the supervised models had to be done

with a smaller dataset, because only 12,408 stations were explicitly categorized as urban, suburban, or rural by the TOAR data providers. The remaining 9,970 stations lacked this specific classification, being reported as 'unknown' or without a designated category. This emphasizes the need for an objective station classification method as described in this paper. For the supervised models, we used 11,408 samples for training and 1,000 for testing, as visualized in (Figure 1. The trained classifier is then employed to predict the characteristics of all 22,378 stations.

To ensure the reliability of our machine learning approach, we manually selected 33 stations with clear decision boundaries and excluded them from the training dataset. These stations were explicitly reported by data providers as urban, suburban, or rural, and we used Google Maps (Mehta et al., 2019) to manually verify and label them. During this process, we observed discrepancies between the labels provided by the data providers and those derived from our manual Google Maps analysis.





Our models will first be evaluated on these 33 stations, with accuracy calculated both against the labels reported by the data
providers and against the manually verified labels from Google Maps (referred to as hand-labeled data).

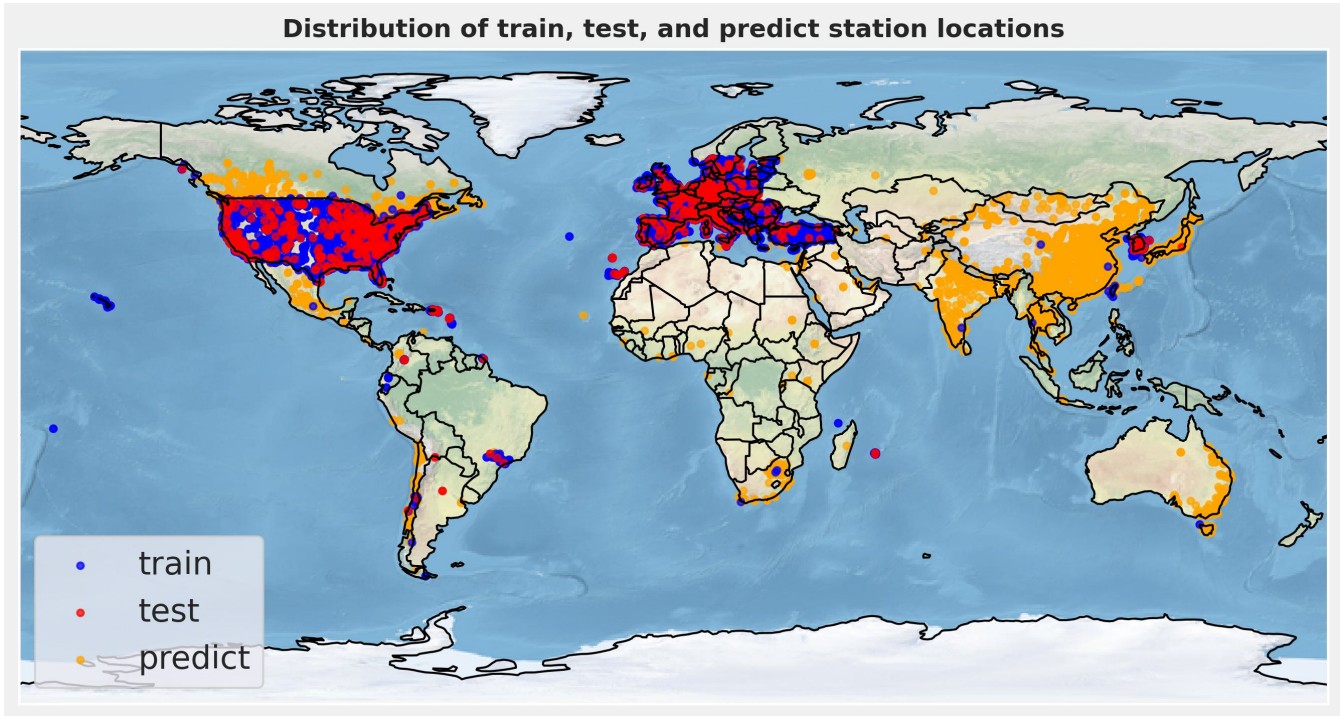

**Figure 1.** Distribution of unlabeled, training, and test data used for the supervised ML models as described in the text

## 2.3 Machine learning algorithms and evaluation methods

This section is devoted to a concise overview of the machine learning techniques employed in this study. Additionally, we
describe the evaluation metrics used to quantify the effectiveness and accuracy of our models.

### 2.3.1 Machine learning algorithms

- **K-means clustering** is a widely-used unsupervised machine learning technique that aims to partition data into k distinct
  groups called clusters Bahmani et al. (2012); Sinaga and Yang (2020); Pelleg et al. (2000). Each data point is assigned to
  the cluster with the nearest centroid. The algorithm seeks to minimize the within-cluster sum of squares, which measures
  the squared distances between data points and their respective centroids. One key requirement of K-means is specifying
  the number of clusters beforehand. We employed the heuristic elbow method to determine the appropriate number of
  clusters for our task. The elbow method is a heuristic technique used to determine the optimal number of clusters in K-
  means clustering. It works by plotting the within-cluster sum of squares (WCSS) as a function of the number of clusters
  and identifying the "elbow" point on the curve, which correspond to the optimal number of clusters (Ketchen and Shook,




1996). In our K-means clustering application, we determine that three clusters are optimal (see Figure 5(a)), which aligns well with our objective of categorizing the stations into three groups: urban, suburban, and rural. However, it is important to note that visual inspection of correlation plots between individual metadata values Figure 1 reveals rather fuzzy boundaries between clusters. This observation aligns with our expectations, given the diverse nature of urban, suburban, and rural locations across different countries, which can vary significantly in terms of industrial development, population density, and degree of urbanization in different countries (Zhang et al., 2024).

- **Random Forest classifier** is a widely-used machine learning algorithm for classification tasks. As an ensemble learning method, it constructs multiple decision trees through bagging during training and outputs the class that is predicted by the majority vote of the individual trees (Breiman, 2001). Known for its robustness, it naturally resists overfitting through random feature selection and typically requires minimal tuning compared to other algorithms. In our implementation, we train the Random Forest classifier with 500 estimators, employing entropy as the optimization criterion. These specific hyperparameters were determined through a grid search. We utilize the RandomForestClassifier from the scikit-learn library (Pedregosa et al., 2011a).

- **The LightGBM (LGBM) classifier** is a supervised machine learning algorithm that utilizes gradient boosting techniques and tree-based learning. It employs histogram-based algorithms and leaf-wise tree growth strategies, which contribute to accelerated training speeds and reduced memory consumption. LightGBM is particularly well-suited for handling large-scale datasets. Its lightweight architecture and optimized algorithm make it a popular choice for tasks requiring both speed and accuracy in prediction (Ke et al., 2017). In our implementation, we train LightGBM with 500 estimators using Python's open-source library 'lightgbm' (Van Rossum et al., 2007).

- **The CatBoost classifier** is a machine learning algorithm that uses gradient boosting on decision trees, specifically designed to handle categorical features seamlessly Prokhorenkova et al. (2018). CatBoost stands for "Categorical Boosting" and automatically handles categorical variables without requiring manual prepossessing. It uses symmetric trees and ordered boosting to prevent overfitting, and often outperforms other methods on datasets with categorical data. This makes it an attractive option for datasets containing both numerical and categorical variables. In our implementation, we employ the open-source CatBoost library and configure the model with 500 estimators to balance performance and computational efficiency.

### 2.3.2 Leveraging Model Uncertainty to Enhance Suburban Classification Accuracy

Considering the inherent subjectivity in defining suburban areas, we refined our prediction methodology as follows: For any given station, if the model's highest probability for either urban or rural classification falls below a threshold (we set this threshold to 50%), and if the second-highest probability corresponds to suburban classification, we interpret this as the model's uncertainty in categorizing the area between rural and suburban, or urban and suburban. In such cases, we classify the station as as suburban. This approach acknowledges the model's indecision and leverages it to better capture the nuanced nature of suburban environments.





### 2.3.3 Evaluation

To evaluate our model's accuracy, we use a separate test dataset of 1,000 samples, shown as red dots in Figure 1. The test dataset consists of samples that were intentionally excluded from both the training phase and the hyperparameter tuning process. This approach ensures that the evaluation metrics provide an unbiased assessment of the model's ability to generalize to new, unseen data, challenging the model in the real-world application scenario. We employed the following evaluation metrics to measure the performance of our machine learning model on this test dataset. We use the following three metrics for the evaluation of our results:

- **Accuracy** measures the ability of the machine learning model to accurately predict the outcome for the given input data. It is measured as the proportion of correct predictions to the total number of predictions made by the model, and given by the following formula:

$$Accuracy = \frac{Number\ of\ Correct\ Predictions}{Total\ Number\ of\ Predictions} \times 100$$

- **Adjusted Rand Index(ARI)** quantifies the similarity between the true cluster assignments and those predicted by the model. It operates by considering all possible pairs of samples and counting how many pairs are assigned to the same or different clusters in both the predicted and true clusters, Chacón and Rastrojo (2023); Chekir et al. (2017). The ARI score ranges from -1.0 to 1.0. A score approaching 1 indicates strong concordance between the true labels and the model's predictions, indicating that many sample pairs are clustered similarly in both clusters. A score near 0 suggests the clustering is comparable to random assignment. A negative score suggests that the predicted clusters frequently disagree with the true clusters, potentially performing worse than random assignment. This implies that sample pairs are often grouped differently in the predicted clusters compared to the true clusters

- **Normalized Mutual Information (NMI)** measures the mutual information between the true clusters of the samples and the clusters assigned by K-means, normalized by the average entropy of the two label sets. It ranges from 0 to 1, where a score close to 1 indicates strong agreement between the true clusters and the K-means clusters. A score of 0 indicates no mutual information between clusters (Kvålseth, 2017).

## 3 Results and discussion

In this section, we present and discuss the results of the different machine learning models that were used for the TOAR station classification task. In the first subsection, we describe and analyze the results from the unsupervised k-means clustering, and in the second subsection, we discuss results from the three supervised methods.





### 3.1 Results for K-means clustering

Figure 1 shows the elbow plot, a heuristic technique used to determine the optimal number of clusters for K-means clustering.

As the gradient of classification accuracy flattens at 3 to 4 clusters, these values for K represent the optimal choices. This result is very encouraging since we want to distinguish 3 different types of stations.

As a first analysis of the K-means clustering, Figure 2(b) shows the K-means predictions evaluated on 33 manually selected and labeled stations, with clear decision boundary from different categories for sanity check of our method. Table 2 presents the accuracy of K-means predictions on these manually labeled stations from the test set, comparing them with the characterization report from the TOAR database.

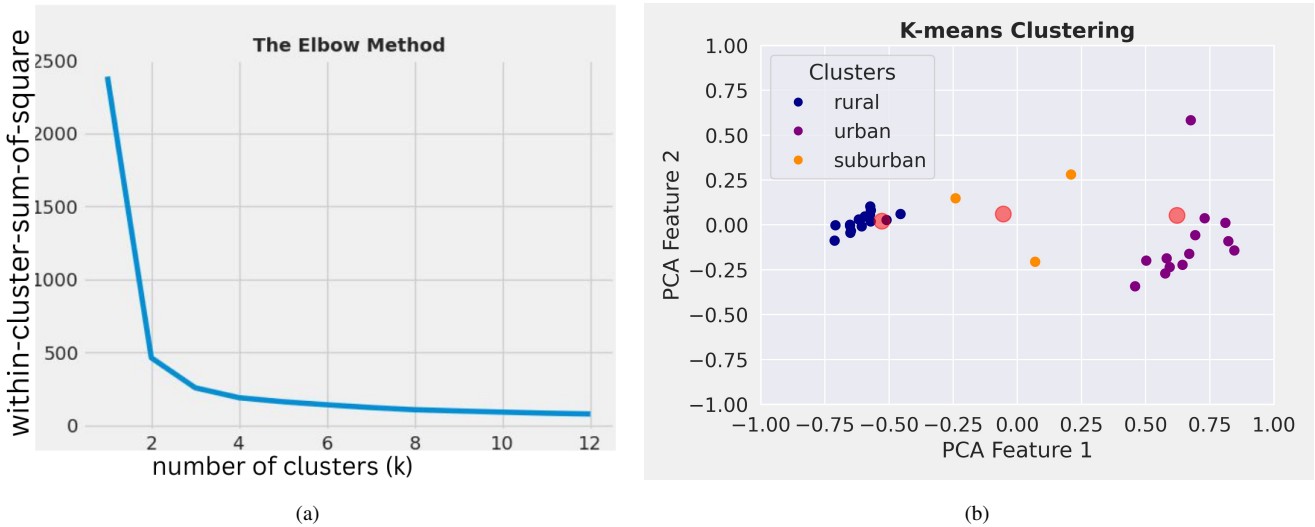

(a)                                                                                        (b)

**Figure 2.** (a) Elbow method to determine the optimal number of clusters for K-means. (b) Different clusters for selected hand-labeled stations, the red points represent the centroid of different clusters. We use Principal Component Analysis (PCA) to project data in 2 dimension for visualization.

**Table 2.** Accuracy, ARI, NMI of the K-means method evaluated on 33 hand-labeled stations and on 1,000 unseen stations labelled by the TOAR data providers.

|  | Accuracy | ARI | NMI |
|---|---|---|---|
| 33 manually selected stations with labels from TOAR data providers | 93.94% | 0.77 | 0.79 |
| 33 manually selected, hand-labelled stations | 87.88% | 0.66 | 0.69 |
| 1,000 unseen stations labelled by TOAR data providers | 65.90% | 0.30 | 0.31 |


Figure 3(a) presents the confusion matrix computed from the K-means prediction and the classes from the TOAR database as ground truth, based on 1,000 unseen test stations (see section 2). This confusion matrix highlights the high accuracy of





K-means in classifying urban and rural stations (70.53% and 71.03%, respectively, see Table 3). However, there are also many instances where reported urban and rural stations are classified as suburban, and vice versa. This misclassification can be
attributed to the subjective nature of defining suburban areas, which often lie between rural and urban regions, or which feature a mix of urban and suburban or rural and suburban characteristics. Additionally, Figure 3(b) visualizes the clusters defined by K-means for the 1,000 unseen test stations. This visual representation clearly illustrates the fuzzy boundaries between the clusters and the noticeable spacing among the three centroids (depicted as red points).

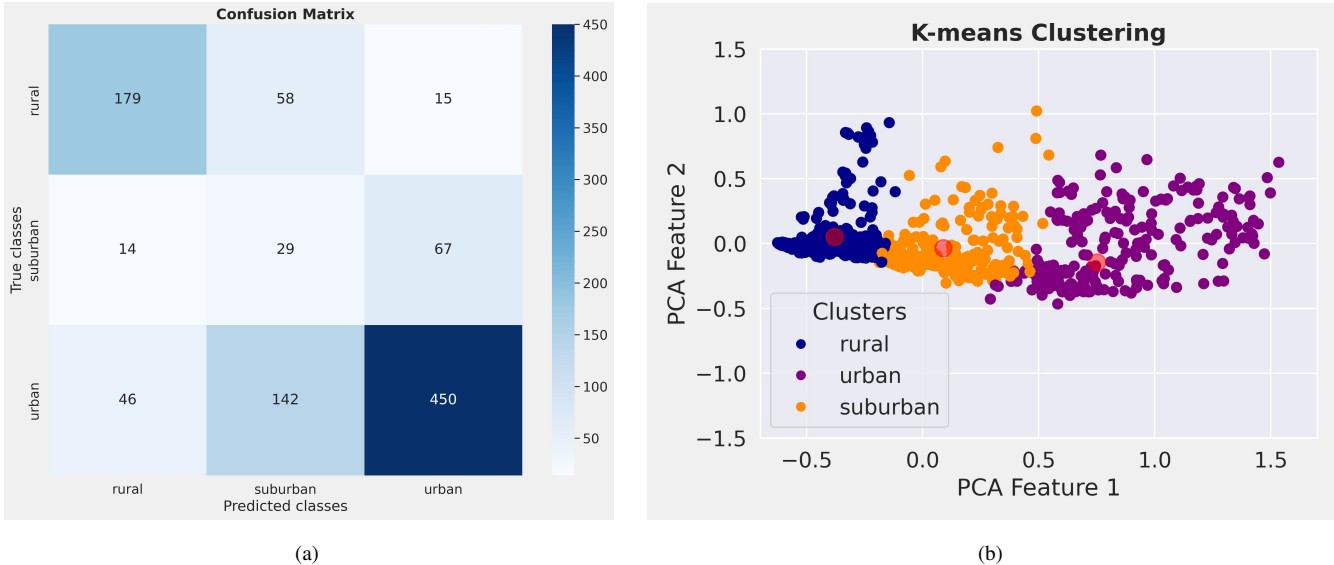

(a)                                                                              (b)

**Figure 3.** (a) Confusion matrix for K-means clustering, evaluated on 1,000 unseen labelled by the TOAR data providers. (b) Cluster visualization of the previously unseen test data. To better visualize the cluster separation, the data is projected into two dimensions using Principal Component Analysis (PCA).

**Table 3.** Detailed accuracy of K-means prediction evaluated on 1,000 unseen test data points, for different classes.

|  | K-means |
|---|---|
| Global Accuracy | 65.90% |
| Accuracy for urban | 70.53% |
| Accuracy for rural | 71.03% |
| Accuracy for suburban | 26.36% |

## 3.2   Results for supervised classifiers

Here, we evaluate the results from the three supervised machine learning classifiers, Random Forest, LightGBM, and CatBoost. Furthermore, the results from the three models were subjected to a robust voting classifier to maximize the classification



accuracy. We observed that the results of all algorithms are quite similar. The models demonstrated exceptional accuracy (> 83%) in predicting urban and rural areas. However, all models struggled with the suburban class, yielding accuracies slightly above 50% (Table 4). As discussed above, the main reason for this lower accuracy can be attributed to the inherent subjective

nature of defining this category. To address this issue, we implemented a strategy that capitalizes on model uncertainty. By adjusting the prediction probability threshold, as detailed in Section 2, we significantly enhanced the accuracy of suburban area classifications as shown in Table 5. Figure 4(a) presents the confusion matrix for the Random Forest classifier and Figure 4(b) visualizes the feature importance. The accuracy of different classifiers is reported in Table 4 and Table 5, which show the results before and after the probability threshold adjustment, respectively. While the overall accuracy remains relatively similar before

and after the adjustment, the probability threshold modification significantly enhances the prediction accuracy for suburban stations, increasing it from a range of 51.74%–54.86% to 58.24%–62.07%. We also note a slight drop in accuracy when classifying urban and rural stations. However, classification performance remains high across all classifiers, with accuracy values exceeding 80%.

        Additionally, we conducted tests on our machine learning models using the manually labeled stations, similar to those used

for K-means evaluation. In this test, we found that the classifiers predict the label report on TOAR by data provider for 33 manually selected stations with 100% accuracy and achieve an 87.88% accuracy for the manual classified stations.

        The implementation of the voting procedure and adjusted probability threshold yielded notable improvements in our classification model. While the enhancements for urban and rural station predictions were modest, the impact on suburban area classification was substantial. This is particularly significant given the inherent challenges in accurately identifying suburban zones. When

compared to the unsupervised K-means clustering method, our supervised approaches demonstrated superior performance across all categories. The contrast was especially pronounced in the classification of suburban areas, where K-means exhibited a markedly low accuracy of just 26.36%. In contrast, our supervised methods achieved significantly higher accuracy rates, underscoring their effectiveness in navigating the complexities of urban-suburban-rural distinctions.

**Table 4.** Accuracy of random forest, LGBM, CatBoost, and voting classifiers before probability threshold adjustment, evaluated on 1,000 test stations.

|  | **Random forest** | **CatBoost** | **LGBM** | **Voting** |
|---|---|---|---|---|
| Global Accuracy | 79.01**%** | 77.50% | 76.20% | 78.70% |
| Accuracy for urban | 91.03**%** | 88.28% | 88.05% | 90.80% |
| Accuracy for rural | 86.64% | 84.12% | 83.03% | 84.84**%** |
| Accuracy for suburban | 53.47% | 54.86**%** | 51.74% | 54.51% |

### 3.3    Discussion

The supervised machine learning approach demonstrates remarkable performance, achieving prediction accuracies > 84% for urban and rural stations when applied to previously unseen test data. This can already be used to accurately predict "urban"





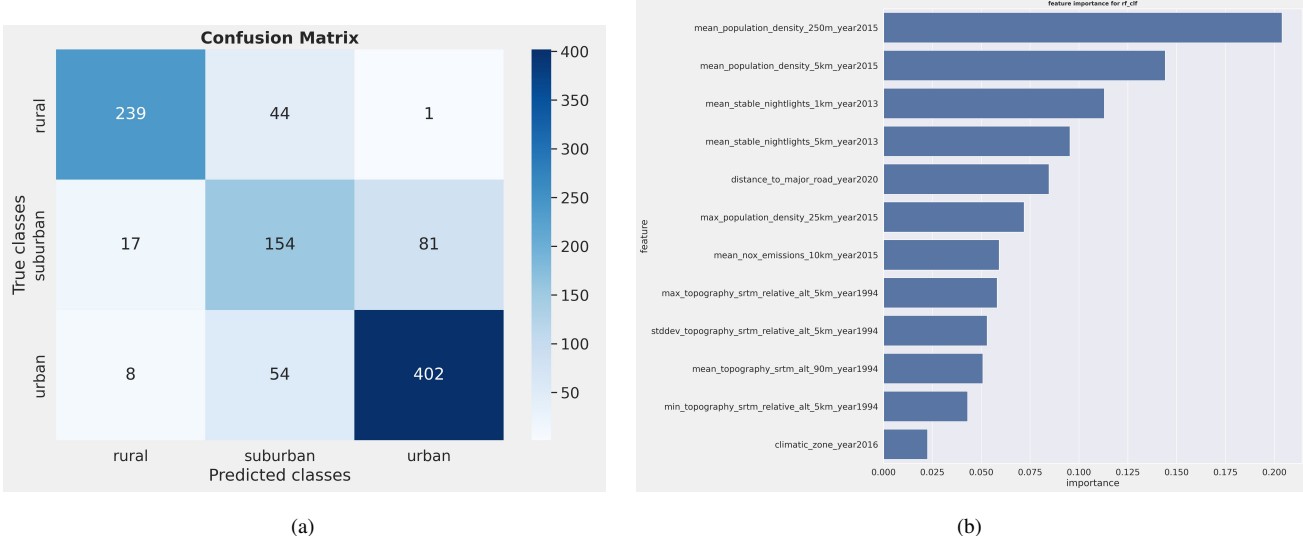

|  | (a) |  | (b) |

**Figure 4.** (a) confusion matrix for random forest classifier, evaluated on 1,000 test data points. (b) the feature importance for random, measuring the contribution of each variable in the classification process

**Table 5.** Accuracy of random forest, LGBM, CatBoost, and voting classifiers after probability threshold adjustment, evaluated on 1,000 test stations.

|  | **Random forest** | **CatBoost** | **LGBM** | **Voting** |
|---|---|---|---|---|
| Global Accuracy | 76.90% | 77.10% | 77.50% | 78.20% |
| Accuracy for urban | 82.93% | 83.15% | 85.56% | 84.90% |
| Accuracy for rural | 80.85% | 81.21% | 82.27% | 82.27% |
| Accuracy for suburban | 62.07% | 62.07% | 58.24% | 62.07% |

and "rural" labels. While the model's performance in identifying suburban areas initially showed slightly lower accuracy, this challenge was effectively addressed through the adjusted probability threshold. Despite the promising results, the classification results are far from perfect. This can be partially attributed to inherent inaccuracies within the dataset itself. To investigate this issue, we conducted a detailed review and manual inspection of the 25 worst misclassifications. For these stations, which are listed in Table 6, we visually inspected the areas around the stations on Google Maps Mehta et al. (2019), using a zoom level of 11 or greater. While 8 of these cases revealed wrong classifications by our best ML model, the model's classification is actually more accurate than the label that was reported by the data providers in 15 cases. In the remaining two cases, neither the reported nor the ML model derived label was correct. In one of these cases, both the reported and ML based label was urban, while the station site is apparently located in a rural area. In the other case, visual inspection would place the station in the suburban class, while the reported category is urban and the ML model classifies the station as rural. However, for stations misclassified by our machine learning model, we observed ambiguous features that could misguide our model. For instance,







surrounding neighborhoods of areas reported as urban but classified as rural by our model exhibited rural characteristics, such as lower population density (which is one of the important features used in the training dataset) and more green space.

**Table 6.** Closer analysis of some misclassified station locations

| latitude | longitude | type of area TOAR | type of area ML | True type of area (From Google Maps) | Winner |
|---|---|---|---|---|---|
| 59.123314 | 11.391136 | rural | urban | urban | ML |
| 34.985670 | -84.375193 | urban | rural | suburban | None |
| 34.710100 | 128.587500 | urban | rural | rural | ML |
| 37.360500 | 128.125600 | urban | rural | rural | ML |
| 41.285533 | -93.583983 | urban | rural | rural | ML |
| 44.785616 | -69.885058 | urban | rural | rural | ML |
| 35.102638 | -85.162194 | urban | rural | urban | TOAR |
| 45.542222 | 9.516389 | urban | suburban | suburban | ML |
| 39.720451 | -94.872693 | urban | suburban | urban | TOAR |
| 48.691113 | 21.286388 | urban | urban | rural | None |
| 35.259502 | -120.644720 | urban | suburban | suburban | ML |
| 37.049115 | -122.019962 | urban | suburban | suburban | ML |
| 41.895812 | -87.607683 | urban | suburban | urban | TOAR |
| 41.193587 | 1.236703 | rural | suburban | rural | TOAR |
| 34.131714 | -109.282309 | rural | suburban | rural | TOAR |
| 33.061150 | -112.052040 | rural | suburban | suburan | ML |
| 49.076550 | 8.406660 | rural | suburban | suburban | ML |
| 36.186210 | -5.380810 | rural | suburban | suburban | ML |
| 45.647310 | 13.854970 | rural | suburban | suburban | ML |
| 43.045269 | -70.713958 | suburban | rural | suburban | TOAR |
| 44.013056 | 12.420000 | suburban | rural | rural | ML |
| 43.186600 | -8.471600 | suburban | rural | suburban | TOAR |
| 36.587027 | -89.546742 | suburban | rural | rural | ML |
| 40.077780 | -6.147220 | suburban | rural | rural | ML |
| 35.498711 | -83.310242 | suburban | rural | suburban | TOAR |

To further lend confidence to our results, we evaluated the 75-percentile statistics of the primary air pollutant concentrations $NO_x$ and $PM_2.5$ from the TOAR database. While data on these species is incomplete, there are sufficient measurements from several regions to yield a meaningful statistic. Figure 5 shows box and whisker plots of the 75-percentiles of $NO_x$ and $PM_2.5$ concentrations aggregated for the year 2015 for the three classes. As expected, urban stations typically show substantially higher concentration levels compared to suburban sites, while rural stations show the lowest concentrations.



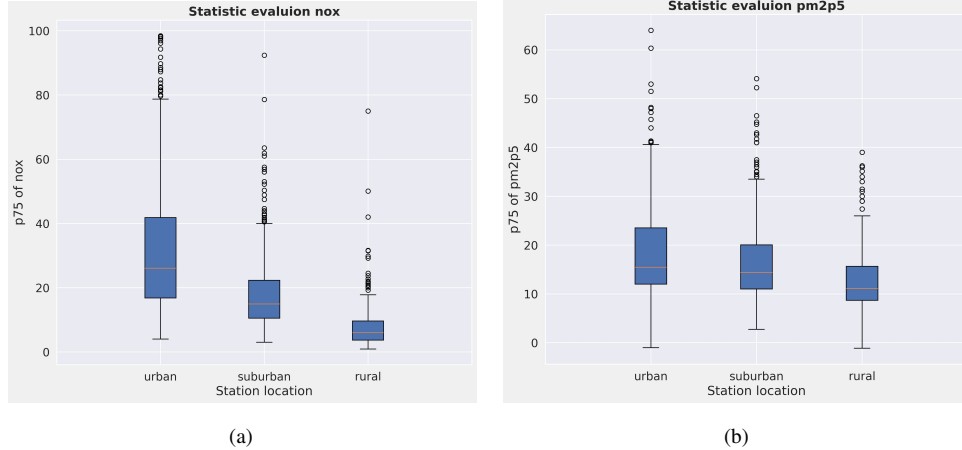

**Figure 5.** Evaluation of the supervised classification with independent data: (a) Box and whisker plot of the 75 percentile of the NOx. (b) Box and whisker plot 75 percentile of the PM2.5

## 4 Conclusion

We investigated the use of machine learning models to objectively characterize station locations for global air quality data analysis. Specifically, we wanted to improve the station classification in the TOAR-I database that was described by Schultz et al. (2017b) and base it on an objective algorithm. As a side-effect we can now explicitly label stations as suburban that were falling between the urban and rural categories in the TOAR-I classification scheme. Our proposed models demonstrate excellent prediction capabilities for urban and rural areas. With the help of an adjusted probability threshold technique, we also obtain meaningful results on the suburban category, inasmuch this category can be described objectively at all. We noticed a limitation for evaluating the accuracy of our method due to obvious misclassification of stations in official databases. As discussed in (Schultz et al., 2017b), such errors can be introduced for various reasons. In some cases, we speculate that these misclassifications actually reflect true landcover changes (e.g., urban development), which have not been updated in the station metadata at the data providers' sites. Manual inspection of worst disagreements revealed that the ML classifier was correct in the majority of cases, where there was disagreement between the reported station type and our ML-derived one.

There is still room for improvement of the methods described here. On the one hand, a larger manual labelling effort using high-resolution EO data, could reduce the number of wrong target labels and reduce the noise in the training data. On the other hand, it may also be possible to employ modern ML methods (e.g., (Szwarcman et al., 2024)) on such high-resolution EO data directly as a specialized land cover classification task. Nevertheless, the new TOAR station classifiers developed in this study provide a clear improvement over the previous method and can be employed in the TOAR-II ozone data analyses that will be reported in the forthcoming assessment papers.





**Code data availability**

All code accompanying this paper is available in our GitLab repository link(Mache et al., 2025) at the following link:

https://gitlab.jsc.fz-juelich.de/esde/toar-public/ml_toar_station_classification/-/tree/develop?ref_type=heads. This repository also contains a copy of the data that is used in this sudy as csv files. The data can also be obtained directly from the TOAR-II database.

**Author contribution**

RKM, SS, and MGS designed the study based on previous work by MGS and SS. RKM, AP, and ML developed the methodology,
RKM implemented the methods and evaluated the results. RKM wrote the major part of the text with contributions from all authors. MGS conducted a final review prior to submission.

**Competing interests**

The authors declare that no competing interests exist.


**Acknowledgements**

The authors are grateful to the EU for funding the IntelliAQ project under grant ERC-AdG-787576. This allowed the buildup of the TOAR-II database. We also greatly appreciate the effort from hundreds of people around the world who established and operate air quality stations, process the data and make the data available to the TOAR initiative. Sebastian Hickman deserves
gratitude for his initial analysis of NOx and PM2.5 data in the TOAR-II database and helpful discussions.





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
