# Peer review of "TOAR-classifier v2: A data-driven classification tool for global air quality stations"

_EGUsphere, 2025_

## Author Comment (AC2)

**Response to Reviewers**

Karim R. Mache

October 2, 2025

We thank both reviewers for their thoughtful comments and constructive feedback. Below we address each point raised.

**Response to Reviewer (RC1) #1**

- Data pre-processing and feature section lacks sufficient details on the preprocessing methodology. This section needs to be revised to provide more clarity. Some examples are listed below.
  - 1. Lines 81-82: Explain the inherent limitations of the NOx dataset which requires applying this normalization. Elaborate on the Box-Cox method as well.
  - 2. Lines 79-80: Provide an excerpt (1-2 lines) about this method for a better understanding of the method.
  - 3. Lines 82-83: Explain the "robust scaler" method.

**Response:** We have expanded the data preprocessing section by providing additional details and outlining the steps more clearly. In this revision, the Box-Cox transformation has been removed. Previously, Box-Cox was applied to the earlier version of the dataset, where the variable NOx contained very small values. The transformation was used in that context to normalize the distribution of NOx. Since this adjustment is no longer necessary for the updated dataset, the Box-Cox step has been excluded.

**Response:**

• Lines 88-89: There is a typo in line 88. It should be "we allocated 21,378 samples for model training...".

**Response:** Done. All numerical values have been thoroughly double-checked and confirmed to be correct.

• Lines 88-94: There are no reasoning for why only 1000 samples were used for testing the ML (both supervised and unsupervised) models. As a rule of thumb, datasets are split 80%-20% into train-test before training ML models to avoid overfitting the training model. Another commonly used approach is train-test-validate split of 70-15-15%. Authors should refer to and base their dataset splitting reasoning on existing literature. The current model in this study might be susceptible to higher misclassifications between urban, suburban and rural labels for unseen datasets.

**Response:** We now consistently apply a train-test split of 85 % and 15 %.

• Lines 88-94: Authors should also include k-fold cross-validation (commonly used is 5-fold CV) while reporting performance metrics for all ML models. The metrics thus derived are more reliable as it ensures a model generalized well to unseen data.

**Response:** We applied tree-based methods that incorporate boosting techniques, which help reduce the risk of overfitting by sequentially improving weak learners and enhancing the model's

generalization ability. Because boosting already provides a built-in mechanism to control overfitting, we did not consider it necessary to include an additional cross-validation step for this stage of the analysis.

- Section 2.3.1: This section very generalized and needs more study-specific information such as:
  - 1. Lines 113-114: How does Figure 5a suggests that 3 clusters are optimal? Is there a scientific technique to arrive at this conclusion, such as a threshold for sum of squares, etc.? Include the explanation.
  - 2. List other hyperparameter values as well for all the ML models in this study. For example: "max\_iter" for k-means cluster; "criterion" and "max\_features" for random forest classifier, etc.
  - 3. Refer to recent studies that have shown similar classification application to justify why these models are specifically chosen in this study. It is done for CatBoost classifier in line 133 but is missing for other models.

**Response:** This section has been revised to include additional details. The choice of three clusters is now justified using an elbow plot, following the standard elbow method, which identifies the point at which increasing the number of clusters yields diminishing improvements in withincluster variance reduction. Regarding the hyperparameters in supervised learning methods, all parameters not explicitly discussed in the text have been retained at their default values as provided by the respective implementations.

• Section 2.3.2: The adjusted threshold probability technique introduced in this study needs some context. Where is the probability derived from? Is it applicable for both supervised and unsupervised learning? Are these probability values an output from the models? If so, explain the steps further that can help retrace the steps in this study for reproducing the results.

**Response:** Yes, these probability values are output from the models.

• Lines 182-185: While it is good to know that the k-means is doing well in labelling urban and rural stations, the main focus of this study is to accurately classify the suburban stations. Therefore, authors should discuss about the classification accuracy of 26.36% for suburban (from Table 3) stations in the text.

**Response:** This is a slight misunderstanding of the reviewer. The main focus of the paper is to produce a robust classification method for the type of area of all stations in the TOAR-II database. The paper only emphasizes suburban sites the most, because these are the most difficult category.

• Line 197: Is the confusion matrix in Figure 4a before or after probability threshold adjustment? Probably mention it in the text as well the figure title.

**Response:** After the probability threshold adjustment, we added more details in the text.

• Line 192: What is the "voting" method? It is mentioned a few times but needs to be clearly defined in methods section.

**Response:** We added a more detailed explanation in the text.

• Figure 5: How does this figure or including NOx and PM2.5 concentrations at all align with the objective of this study? The trend presented in Figure 5 is generally true for both pollutants, but it is not pertinent here. The authors should provide strong reasoning if they still feel the need to include it.

**Response:** Reviewer #2 liked it and we also believe that this additional analysis using an independent dataset lends further confidence in our classification method.

**Response to Reviewer (RC2) #2**

• **Comment:** In the Introduction, define the classes upfront (plain language). Add a short description of urban / suburban / rural in this context, and link it to the features listed in Table 1 so readers understand how these relate to the categories.

Response: Done.

• **Comment:** In the Introduction, when outlining the problem and motivating the study in the last paragraph, consider mentioning that almost 10,000 of the 24,000 stations have no label, which calls for an automated approach.

**Response:** Done. All numerical values have been thoroughly double-checked and confirmed to be correct.

• Comment: Inconsistency claim (L36/37). You state that TOAR labels are inconsistent/errorprone. Please provide a reference or a concrete rationale/example for this statement. (You later discuss misalignment's; connect those to this claim.)

**Response:** Done. Reference to Tapia et al., 2016 added.

• **Comment:** Section 2.2: Provide information on the distribution of TOAR labels in the dataset. How many cases are in the suburban class? Are the class distributions approximately balanced?

**Response:** We have included the distributions of the entire dataset and of the labeled data in Figure 1

• **Comment:** L88: Check the numbers. If you remove 1,000 cases, you should end up with 21,378, I believe.

**Response:** All numbers have been double-checked and updated. The analysis has been repeated with a consistent training-test split of 85 % versus 15 %.

• Comment: L88–94: You have 12,408 stations with labels, but you cluster and predict the entire dataset of 22,378 stations. What is being done with the 10,000 stations that have no label? Is the predicted classification of these stations used for something in the manuscript? Please make this more obvious.

**Response:** The purpose of the paper is to provide a coherent and consistent labelling for all stations in the TOAR-II database, including the previously unlabeled stations. As shown in the discussion, the trained model also provides more accurate labels in some cases than the manually assigned labels by the data providers.

• Comment: L95–99: This is a very important paragraph, as it addresses the reliability of the TOAR labels. Please report explicitly the level of agreement between TOAR labels and the manual approach. What was the distribution of labels in each classification approach, and how often did the suburban category agree/disagree? Did one or several of the authors (independently?) perform the manual classification? This information is crucial to assess the reliability of the reference labels. What do you mean by "clear decision boundaries"? Did you pick particularly easy stations for the manual labeling exercise? Is the proportion of agreement between TOAR labels and manual labels higher than for the 25 worst predictions (Table 6) (if I counted correctly, 8 out of 25 agree)?

**Response:** This was only intended to be a robustness and plausibility check. A more systematic investigation of potential flaws of the reported station labels and our manual approach is beyond the scope of this analysis.

• **Comment:** Figure 1: It looks as if you primarily predict stations outside the U.S. and Europe. Are the orange dots the 1,000 stations used for testing? Or are these the 9,970 stations lacking classification?

**Response:** Orange represent all unlabeled stations. Note that reported classification labels were only provided by a subset of data providers, primarily from Europe, the U.S., and South Korea.

• Comment: Section 2.3.2, L184, L194 – Adjusted probability threshold: The proposed "uncertainty-based" rule for suburban classification is essentially heuristic. A more standard approach would be threshold optimization via grid search on a validation set, maximizing e.g. macro-F1 or balanced accuracy. I am not convinced that the chosen approach truly acknowledges the models' uncertainty between these classes. Instead, the approach simply assigns the middle class. Wouldn't't it have been more transparent to label these as uncertain cases (e.g., rural–suburban, suburban–urban)? This would have allowed the quality of predictions for "in-between" cases to be assessed separately from those where labels and predictions should be more reliable.

**Response:** Done, we implement a grid-search adjusted probability threshold, maximizing the macro-F1

• Comment: Section 2.3.3 – Performance metrics: The analysis relies heavily on per-class accuracy. While accuracy is intuitive, it is not an appropriate sole metric under class imbalance and noisy labels (for instance when describing "global accuracy"). In this case, urban and rural likely dominate (numbers are missing), while suburban is both smaller and potentially much noisier. More suitable and widely accepted measures include per-class precision, recall, F1, macro-F1, and balanced accuracy.

**Response:** Done, we add the all necessaries evaluation metrics.

• **Comment:** Figures 2 and 3: Please introduce PCA as a method to visualize results in the Methods section.

**Response:** Since PCA is not used explicitly for our classification task but only for visualization, we provide an explanation in the text along with a reference.

• **Comment:** Table 2 – interpretation of TOAR vs manual labels: How do you explain that models trained and tested with TOAR labels tend to achieve better performance statistics than with manual labels? Could this indicate that the manually derived labels are themselves noisy, or that they reflect a different conceptual perspective? A more explicit discussion of this is needed.

**Response:** We believe that 30 data points are insufficient to draw definitive conclusions, as this may be influenced by the subjective definition of station categories and noise in the training data. However, the tree-based model used is robust to such noisy data..

• **Comment:** L191, L207, Tables 4 and 5: The so-called "voting procedure" is not well explained. From the text it appears to be simple majority voting across three classifiers. If so, please state this explicitly in the Methods section, and later discuss whether the gain is meaningful relative to the best-performing individual model.

**Response:** We added a more detailed explanation on the text.

• Comment: L220–229: How did you decide on the 25 worst misclassifications? What is the gold standard for this analysis? Is it the manual labels? From Table 2, I wonder whether the manual labels are more reliable than the TOAR labels. A fair comparison, in my view, would involve both a comparison against TOAR and against the manual labels. Discuss that TOAR and manual labels also showed low agreement in this case.

**Response:** The worst misclassifications have been replaced by 30 randomly selected cases of misclassification and this has been updated in the text.

• Comment: L230: Use of pollutant concentrations. The evaluation with NOx and PM2.5 is a useful plausibility check. However, the choice of the 75th percentile is not self-evident. Please

justify why the 75th percentile was chosen (e.g., is this a standard in this field or intended to capture robustly elevated concentrations while avoiding sensitivity to extremes).

Response: Done

• **Comment:** Figures: I suggest removing the grey background. Check with journal guidelines, but I believe (a) and (b) should be shown above the plot (beside the plot title, if the journal allows the latter).

**Response:** We have removed the gray background from all figures.

• **Comment:** Discussion: Assuming that the classification approach is supposed to be implemented in the database, I suggest summarizing key facts for users of these labels. What works, what does not? Following your analysis, do you see any points that could be improved regarding the definition of the three classes?

**Response:** The conclusion section has been slightly extended in this spirit.

**General Revisions**

In addition to the specific points raised by the reviewers, we have also:

- Fixed minor grammatical errors throughout the manuscript and slightly revised the wording in a few places to improve clarity
- Improved the clarity of several figures.
- Add more details for better understanding of our manuscript.
- · Added additional references as suggested

We believe these revisions have significantly improved the quality of our manuscript.